# Randomized Phase II Study with Cetuximab in Combination with 5-FU and Cisplatin or Carboplatin Vs. Cetuximab in Combination with Paclitaxel and Carboplatin for Treatment of Patients with Relapsed or Metastatic Squamous Cell Carcinoma of the Head and Neck (CETMET Trial)

**DOI:** 10.3390/cancers12113110

**Published:** 2020-10-24

**Authors:** Georgios Tsakonas, Lena Specht, Claus Andrup Kristensen, Maria Herlestam Calero Moreno, Hedda Haugen Cange, Karin Soderstrom, Signe Friesland

**Affiliations:** 1Theme Cancer, Medical Unit Head&Neck, Lung and Skin Cancer, Karolinska University Hospital, 17176 Stockholm, Sweden; maria.herlestam-calero-moreno@sll.se (M.H.C.M.); signe.friesland@sll.se (S.F.); 2Department of Oncology-Pathology, Karolinska Institutet, 17176 Stockholm, Sweden; 3Rigshospitalet, Section for Head&Neck and Thoracic Oncology, Department of Oncology, University of Copenhagen, 2100 Copenhagen, Denmark; lena.specht@regionh.dk (L.S.); claus.andrup.kristensen@regionh.dk (C.A.K.); 4Department of Oncology, Sahlgrenska University Hospital, 41345 Goteborg, Sweden; hedda.haugen@vgregion.se; 5Department of Oncology, Norrlands University Hospital, 90185 Umea, Sweden; karin.soderstrom@onkologi.umu.se

**Keywords:** relapsed/metastatic SCCHN, cetuximab, carboplatin/paclitaxel, cisplatin/5-FU, first-line therapy

## Abstract

**Simple Summary:**

The purpose of the CET-MET trial was to find a new platinum- based chemotherapy regimen in combination with cetuximab for relapsed or metastatic squamous cell carcinoma of the head and neck (RM- SCCHN), that would achieve an equivalent PFS with standard cetuximab and 5-FU/platinum-based chemotherapy (EXTREME regimen), albeit with less toxicity. RM-SCCHN is a disease which affects patients with severe comorbidity and unhealthy life styles, rendering it difficult to treat with toxic regimens such as the EXTREME trial regimen. Immune checkpoint inhibition (ICI) with/or without the addition of chemotherapy has recently been introduced as a first- line treatment option for RM-SCCHN. However, these new treatment options will not be suitable for all patients. The experimental arm of this trial with Cetuximab and paclitaxel/carboplatin is easier to administer and perhaps more beneficial to combine with ICIs due to its favorable toxicity profile and the potential immunomodulatory effects of taxanes.

**Abstract:**

Background: Platinum-based chemotherapy with cetuximab is the standard of care for relapsed or metastatic squamous cell carcinoma of the head and neck (SCCHN). The aim of this trial was to investigate whether cetuximab and paclitaxel/carboplatin can achieve similar progression-free survival (PFS) with standard cetuximab and 5-FU/platinum-based chemotherapy. Standard chemotherapy treatment for SCCHN is related to severe toxicity and new, less toxic regimens are needed. Methods: In this multicentre, randomized, controlled, phase 2 trial, 85 patients with relapsed or metastatic SCCHN were randomized in a 1:1 ratio to cetuximab and 5-FU/cisplatin or carboplatin (arm A) vs. cetuximab and paclitaxel/carboplatin (arm B). Eligibility criteria included age ≥18 years, Eastern Cooperative Oncology Group (ECOG) performance status (PS) of 0–1, and adequate organ functions. The primary endpoint was to investigate whether PFS in arm B is significantly worse than PFS in arm A. Results: Median PFS in arm A was 4.37 months (95% CI: 2.9–5.9 m) and 6.5 months (95% CI: 4.8–8.2 m) in arm B, (*p* = 0.064). Median overall survival (OS) was 8.4 months (95% CI: 5.3–11.5 m) in arm A and 10.2 months (95% CI: 5.4–15 m) in arm B, (HR = 0.71; 95% CI: 0.43–1.16). PFS HR for arm B was not significantly worse than arm A (HR = 0.65; 95% CI: 0.41–1.03). Adverse events ≥ grade 3 were more frequent in arm A than arm B (60% vs. 40%; *p* = 0.034). Conclusion: Cetuximab and paclitaxel/carboplatin was found to have similar efficacy and less toxicity compared to cetuximab and 5-FU/cisplatin or carboplatin. The experimental arm is easier to administer rendering it a favorable alternative to standard therapy.

## 1. Introduction

Relapsed or metastatic squamous cell carcinoma of the head and neck (RM-SCCHN) has a dismal prognosis. Treatment strategies that can be employed vary from potentially curative salvage surgery or re-irradiation to mostly palliative systemic therapies including best supportive care. Chemotherapy—in single or multiple drug combinations—has provided a median overall survival (OS) of 6 to 9 months in the first-line setting [1,2,3,4]. Platinum-based chemotherapy regimens have long been the standard treatment for RM-SCCHN, with cisplatin showing higher response rates (RRs) than carboplatin, without any difference in OS [2,5]. Platinum compounds have been used in combination with 5-FU or taxanes with similar OS rates and RRs [6]. For patients who relapse after first-line platinum-based chemotherapy, prognosis is poor with an RR 4–14% and OS 4.3–6.7 months [7].

Cetuximab, a monoclonal antibody targeting the epidermal growth factor receptor (EGFR), has shown clinical activity when used in addition to radiotherapy for locally advanced SCCHN and also as monotherapy for RM-SCCHN patients who have failed platinum-based chemotherapy [8,9]. Cetuximab with platinum/5-FU was compared to platinum/5-FU in the first-line setting in the EXTREME trial, showing better efficacy with a median OS of 10.1 months vs. 7.4 months [10]. Until recently, the EXTREME regimen has served as the standard first-line treatment for RM-SCCHN, though is not widely applicable in clinical praxis, due to its toxicity profile. The TPEx regimen with docetaxel, cisplatin and cetuximab, followed by cetuximab maintenance, seems to be another reasonable option in the first-line setting. This regimen has lower toxicity than the EXTREME regimen, but does not result in better OS [11].

Immunotherapy has recently emerged as a treatment option for RM-SCCHN in the first- and second-line setting [12,13]. The KEYNOTE-048 trial, presented at European Society of Medical Oncology (ESMO) 2018 congress, has demonstrated anOS benefit of the PD-1 antibody pembrolizumab in first-line, both as monotherapy (in the subgroups of tumors with PD-L1 combined positive score (CPS) > 1, and CPS > 20) and in combination with platinum chemotherapy (irrespective CPS score) compared to standard platinum/5-FU + cetuximab chemotherapy. The results of the KEYNOTE-048 trial are expected to change the treatment guidelines for RM-SCCHN, although there are concerns regarding the toxicity of the immunotherapy/chemotherapy combination, due to high rates of grade 3 and 4 adverse events, as well as high rates of treatment discontinuation [13].

A randomized phase 2 trial was designed in order to investigate whether cetuximab and paclitaxel/carboplatin can achieve similar progression-free survival (PFS) with less toxicity compared to the EXTREME regimen.

## 2. Results

### 2.1. Patient Characteristics

A total of 85 patients were screened from 1 November 2011 to 1 March 2017. Three patients were excluded after randomization, two patients because of deterioration before treatment start and one patient because the inclusion criteria were not fulfilled. All patients that were screened were included in the intention to treat (ITT) analysis (Figure 1). Baseline characteristics including age, gender, Tumour Node Metastasis (TNM) staging at initial diagnosis, localization, Human Papilloma Virus (HPV) status, smoking status and performance status (PS) were well balanced between the two treatment arms (Table 1). The median age for the whole study population was 60.9 years, with a male predominance (69.4%). The vast majority of the study population had non-hypopharyngeal tumors which were HPV-negative). In total, 23 (54.8%) patients received cisplatin, 17 (40.5%) patients received carboplatin and 2 (4.7%) patients did not receive any treatment as mentioned above. A total of 18 patients in arm A (43%) vs. 26 patients in arm B (61%) received all 6 planned cycles of chemotherapy. All patients included in the trial had a complete response to prior treatment which was given with a curative intention. All prior treatments were well balanced between the 2 study arms. All radiotherapy treated patients received doses up to 65 + Gy (the vast majority 68 Gy) and a few patients received neoadjuvant treatment (Table 1). Six patients (14%) in arm A and three patients (6.7%) in arm B received concomitant radiotherapy with cetuximab, whereas 18 patients (40%) in arm A and 15 patients (33.3%) in arm B received concomitant radiotherapy with cisplatin (*p* = 0.53).

-Arm A (every 21 days): D1 cisplatin 100 mg/m^2^ or carboplatin 5 AUC, D1-4 5-FU 1000 mg/m^2^, D1 cetuximab 400 mg/m^2^ (first administration and then 250 mg weekly)

-Arm B (every 21 days)**:** D1 carboplatin 5 AUC, D1 paclitaxel 175 mg/m^2^, D1 cetuximab 400 mg/m^2^ (first administration and then 250 mg weekly)

### 2.2. Response and Survival Analyses

The cut-off date for data analysis was 19 December 2017. Median duration of follow-up for PFS was 4.32 months for arm A, and 6.48 months for arm B (5.7 months for both arms). Median duration of follow-up for OS was 8.4 months for arm A, and 10.2 months for arm B (9.3 months for both arms). Median overall survival (OS) was 8.4 months (95% CI: 5.3–11.5 m) in arm A and 10.2 months (95% CI: 5.4–15 m) in arm B, (HR = 0.71; 95% CI: 0.43–1.16) (Figure 2A). Median PFS in arm A was 4.37 months (95% CI: 2.9–5.9 m) and 6.5 months (95% CI: 4.8–8.2 m) in arm B, (*p* = 0.064, non-stratified log rank test) (Figure 2B). *p*-value for the log-rank test between treatment groups for PFS stratified on previous chemotherapy and tumor sites was 0.0891. PFS HR for arm B was 0.65 (95% CI: 0.41–1.03), therefore not significantly worse than arm A and with a strong trend to superiority towards arm A.

Median Time to Treatment Failure (TTF) for arm A was 3.95 m (95% CI: 1.92–5.96), whereas for arm B it was 5.87 m (95% CI: 3.02–8.69) (Figure 2C). TTF was significantly longer for arm B—HR = 0.63 (95% CI: 0.40–0.99). Best overall response was similar in both arms (20 vs. 22 patients). However, there were proportionally more patients with a complete response (CR) in arm B vs. arm A: 7 vs. twopatients, partial response (PR): 13 vs. 20, resulting in a statistically significant difference between the 2 arms (*p* = 0.041). Median duration of response for arm A was 6.44 m (95% CI: 4.38–8.51) and 7.37 m (95% CI: 4.84–9.89) for arm B, HR = 0.51 (95% CI: 0.25–1.06) (Figure 2D).

The patient and tumor characteristics were well balanced between the two treatment arms besides gender (Table 1); therefore, we corrected the univariate analyses for OS, PFS and TTF only for that variable. There was also a relative imbalance between the two study arms regarding HPV status. HPV status was not included in the multivariate analysis due to the large amount of missing data in arm A compared to arm B (16.7 vs. 2.3%), rendering it a suboptimal variable for further testing. There was not any significant difference in HRs for PFS, OS and TTF after the multivariate analysis (Table 2).

### 2.3. Toxicity

There was a statistically significant difference in adverse events (AEs) between the two treatment arms. Adverse events ≥ grade 3 were more frequent in arm A than in arm B (60 vs. 40%; *p* = 0.034). Hematologic toxicity was the most frequent AE in both arms, but with more events in arm A. Gingivostomatitis, oral mucositis, diarrhea, heart failure and thromboembolic events were not observed in the experimental arm. Four cases with grade 5 adverse events were reported—two in the experimental arm (sepsis and bleeding from oral cavity) and two in the control arm (thromboembolism and colon perforation). (Table 3).

## 3. Discussion

The aim of the present randomized phase II study was to find a regimen with a comparable efficacy to the EXTREME regimen but with better tolerability and therefore with a potentially more widespread clinical usefulness. The results of the study can be said to have met this primary endpoint, since the PFS of study arm B was not significantly inferior to the EXTREME arm, and less serious toxicity was observed in the experimental arm.

RM-SCCHN is a disease with limited treatment options. Tolerability to tested regimens has been low in this frail patient population. The rationale for combination regimens rather than single agent therapy has been based on greater response rates and higher rates of locoregional control. However, such combination chemotherapy increases toxicity significantly and has failed to improve overall survival. For several decades before the introduction of the EXTREME regimen, no studied combinations of cytotoxic drugs or targeted therapies yielded any increase in survival of these patients [1,2,3,4,5,6]. The increase in OS from 7.4 to 10.1 months and progression-free survival from 3.3 to 5.6 months in the EXTREME trial was rather modest and was accomplished in a subset of patients with good performance status. The EXTREME trial regimen (arm A in our trial) has been the gold standard for first-line treatment in many years but has limited applicability in the clinical setting due to its toxicity [10,14]. The majority of patients with RM-SCCHN cannot tolerate this regimen and new, less toxic treatments are needed in daily clinical practice. The slow accrual of patients in our study highlights this clinical problem, since it was difficult to find eligible patients who could be randomized between the two treatment arms, mainly due to inclusion criterium with PS 0-1. This was the reason for the early termination of our study which resulted in less statistical power regarding the primary endpoint of PFS. The TPEx regimen is a less toxic alternative to EXTREME. However the TPEx regimen contains cisplatin which is more toxic than carboplatin, and failed to show any OS benefit [11].

The experimental arm (arm B) proved to be less toxic and had a similar efficacy to the EXTREME regimen. PFS in arm B was not significantly worse than arm A, and a strong trend towardssuperiority was observed for arm B compared to arm A. This positive PFS result is, however, supported by low statistical power, since the study was prematurely terminated due to slow accrual. This slow accrual was related to the strict inclusion criteria, especially regarding PS 0-1. RM-SCCHN is a disease which affects patients with severe comorbidity and unhealthy lifestyles, rendering it difficult to treat with toxic regimens. The inclusion of such patients in randomized clinical trials is a challenging issue. TTF was significantly better for the experimental arm (B) in a direct superiority comparison. This result is highly relevant since TTF is an endpoint greatly influenced by treatment-related toxicity. In the present study, this correlates with the favorable toxicity profile in arm B and resulted in lower rates of treatment discontinuation in the experimental arm. There was a statistically significant difference in grade ≥ 3 adverse events between the treatment arms in favor of the study arm. The frequency of any event at all was also higher in the control arm. It seems that the combination of 5-FU and cisplatin—not only cisplatin—is responsible for the difference in toxicity between the two treatment arms since a large number of patients (40.5%) received carboplatin/5-FU in arm A. There were more patients who completed all six planned cycles of chemotherapy in arm B (61%) compared to arm A (43%), something which is indicative of the severe toxicity related to the control arm. The advantageous toxicity profile of the experimental arm could partially be explained by the lack of side-effects from infusional 5-FU, such as mucositis, diarrhea and nausea. Less episodes of infections and thromboembolic events were observed in arm B, something which is of great importance for treatment decisions in RM-SCCHN patients. The experimental arm consists of a regimen which is easier to administer and therefore more convenient for RM-SCCHN patients; carboplatin can be administered in a shorter period of time compared to cisplatin, and paclitaxel is infused only on day 1 of each cycle, without the need for an infusional pump that the patient has to carry from day 1 to day 4 of each cycle. All other endpoints had a positive trend towards favoring the experimental arm, though no statistically significant differences were observed, except for the OR rate which was superior in the experimental arm, due to a proportionally larger number of patients with CR. It is difficult to measure tumors of the head and neck, especially when response is determined for assessable disease in patients that may be heavily pre-treated. However, the criteria for a complete response in this region are reasonably easy to adhere to, leading to a good reproducibility. In this study, there were seven patients in arm B and three in arm A with complete remissions.

Immune checkpoint inhibition with PD-1 antibodies has recently been introduced as a treatment option for RM-SCCHN. Two randomized phase III clinical trials, Checkmate-141 and Keynote-040, performed on patients with platinum-refractory RM-HSCCN, demonstrated an improved overall survival in RM-HSCCN with nivolumab and pembrolizumab, respectively [12,15]. The newly presented results of the KEYNOTE-048 study (pembrolizumab single or in combination with chemotherapy vs. the EXTREME regimen) in the first-line setting are really promising and are going to change the treatment algorithm of RM-SCCHN. However, these new treatment options will not be suitable for all patients. Pembrolizumab monotherapy, which is less toxic compared to the EXTREME regimen, seems to be a good treatment option only in patients expressing PDL-1, especially in the subgroup of patients with CPS > 20. Pembrolizumab alone showed a more profound OS benefit vs. cetuximab with chemotherapy in the CPS of 20 or more of the population (median 14.9 months vs. 10.7 months, HR = 0.61; 95% CI 0.45–0.83) compared to the CPS of 1 or more of the population (12.3 vs. 10.3, HR = 0.78; 95% CI: 0.64–0.96). Combination therapy with pembrolizumab and standard chemotherapy will be a suitable regimen in patients who are PDL-1-negative and eligible to the EXTREME regimen and is going to most likely substitute standard treatment with certain toxicity issues [13]. The problem in daily clinical practice is that not all patients with RM-SCCHN can tolerate these toxic regimens, the overall response rate with PD-1 monotherapy is low and the cost of treatment is high. Moreover, most patients will ultimately progress on these agents and reliable predictive biomarkers for responses are awaited.

Cetuximab and paclitaxel/carboplatin, followed by cetuximab maintenance (arm B) can be a reasonable first-line treatment option for patients not expressing PDL-1, and who cannot tolerate the EXTREME regimen or combination therapy with pembrolizumab plus chemotherapy. The experimental arm in our trial is most probably less toxic than the TPEx regimen which contains cisplatin, rendering it a better treatment option for patients who cannot tolerate cisplatin. Paclitaxel/carboplatin maybe is a better choice of chemotherapy than platinum/5-FU when combined with immunotherapy due to its favorable toxicity profile and the immunomodulatory effects of taxanes regarding the activation and accrual of intratumoral T effector cells and NK cells [16]. Taxanes have been successfully combined with immunotherapy in both squamous non-small cell lung cancer and triple-negative breast cancer patients [17,18]. This hypothesis should be tested in a randomized setting for RM-SCCHN. If immunotherapy (with or without chemotherapy) is given in the first-line setting, then the optimal second-line treatment is controversial. The use of cetuximab in combination with platinum-based chemotherapy could be a reasonable treatment option for these patients, and less toxic treatments than the EXTREME regimen may be preferable in the clinical setting of second-line therapy of RM-SCCHN.

## 4. Materials and Methods

### 4.1. Patient Eligibility

This multicentre randomized controlled phase 2 trial was conducted at 3 University Hospitals in Sweden and 1 in Denmark. Randomized controlled trials registration number: Eudract No: 2010-022924-57. Eligible patients were histologically or cytologically confirmed RM-SCCHN which was not suitable for local treatment alone. Eligibility criteria included age ≥18 years, Eastern Cooperative Oncology Group (ECOG) performance status (PS) of 0–1, adequate organ functions and at least one dimensionally measurable lesion either by CT scan or MRI or physical examination according to RECIST 1.1. Key exclusion criteria were: age > 75 years, nasopharyngeal cancer and cancer of the paranasal sinuses, inability to follow the treatment and evaluation schedule, active infections or any other serious underlying medical condition, inadequate organ functions, clinically relevant neuropathy, previously treated for RM-SCCHN (except radiotherapy for previously treated relapse if terminated > 3 months before start of treatment), previously treated with cetuximab, cisplatin/carboplatin, 5-FU or taxanes for locally advanced SCCHN within 3 months before study entry.

### 4.2. Study Design

Patients were randomly allocated to receive treatment with either cetuximab in combination with 5FU and carboplatin or cisplatin (arm A) for a maximum 6 cycles, or cetuximab and paclitaxel/carboplatin (arm B) for a maximum of 6 cycles. Randomization occurred centrally on a 1:1 basis. Doses and dose intervals were as follows: Arm A—initial cetuximab dose of 400 mg/m^2^, followed by weekly doses of 250 mg/m^2^. 5-FU 1000 mg/m^2^ Day 1–4, every 3 weeks, cisplatin 100 mg/m^2^ day 1, every 3 weeks, carboplatin AUC 5 day 1, every 3 weeks. Arm B—initial cetuximab dose of 400 mg/m^2^, followed by weekly doses of 250 mg/m^2^, paclitaxel 175 mg/m^2^ day1, every 3 weeks, carboplatin AUC 5 day 1, every 3 weeks. Patients without disease progression continued with cetuximab maintenance 500 mg/m^2^ every second week until progression or toxicity. Treatment according to study protocol started within 2 weeks after randomization and randomization was stratified for tumor localization (hypopharynx vs. non-hypopharynx tumors). Dose delays or dose modifications were specified for hematologic, gastrointestinal, neurologic and renal toxicities. For all grade 3–4 toxicities, treatment was withheld until complete resolution of the toxicity and then resumed with dose modification according to the treatment protocol.

The study was undertaken in accordance with the Declaration of Helsinki and Good Clinical Practice Guidelines. The protocol was approved by the ethics review board of every participating institution and all patients provided written informed consent.

### 4.3. Assessment

MRI or CT (scanning from the base of skull to upper abdomen) was performed at baseline, every 3 months during the first 2 years of follow up and thereafter every 6 months for 3 more years, or until progressive disease (PD) according to RECIST version 1.1, death or study withdrawal. Radiographic controls were performed earlier than scheduled if clinically indicated. If PET/CT was performed at baseline and after cycle 2, the first tumor evaluation was replaced by PET/CT. The following tumor assessment was continued with MRI or CT. The primary endpoint was PFS (non-inferiority design). Secondary endpoints were best overall response (BOR), duration of response (DOR), time to treatment failure (TTF), overall survival (OS) and safety (using The National Cancer Institute Common Terminology Criteria for Adverse Events version 4.0).

### 4.4. Statistical Analysis

In the EXTREME trial a median PFS time of 5.6 months was observed in patients treated with cetuximab + 5-FU + cisplatin/carboplatin (arm A in this trial) [10]. If a total number of 110 events (progression or death) would be observed (the initial design of this study), a log-rank test would have 80% power to reject the null hypothesis of inferiority (a hazard ratio of 1.5 or greater) with a one-sided significance level of 0.1, when the true hazard ratio is 1. The following assumptions were made for the initial sample size calculation: the hazard rate in arm A is 0.1238/month (equivalent to a median PFS rate of 5.6 months), the accrual period is 24 months, the maximum follow-up time is 50 months and the dropout rate per month is 1% in each arm. Due to slow accrual, our study was terminated earlier and a new power calculation was made. This decision was taken due to concerns about trial futility in case of study continuation until the initial number of events had taken place and also due to the fact that the initial randomization criteria were not planned to be altered. A non-inferiority log-rank test with an overall sample size of 80 subjects (40 in the reference group and 40 in the treatment group) achieves 67.7% power at a 0.100 significance level to detect an equivalence hazard ratio of 1.50 when the actual hazard ratio is an equivalence hazard ratio of 1.00 and the reference group hazard rate is 0.1238.

### 4.5. Statistical Methods

In total, 85 patients with relapsed or metastatic SCCHN were randomized in a 1:1 ratio to cetuximab and 5-FU/cisplatin or carboplatin (arm A, *n* = 42) vs. cetuximab and paclitaxel/carboplatin (arm B, *n* = 43). The intention-to-treat (ITT) cohort included all patients who were randomized to a study treatment, whereby patients were analyzed according to the group to which they were randomized. The PFS curves in both treatment arms were compared using a log rank test at a one-sided significance level of 0.1. The log rank test was stratified by the factors considered during randomization (previous chemotherapy and tumor site). In addition, the primary and the secondary endpoints were also analyzed using the log-rank without any stratification. Univariate Cox proportional hazard methods were used to estimate the hazard ratio (and corresponding 95% CI) between both treatment arms. Furthermore, a multivariable Cox proportional hazard method was used to investigate the impact of possible prognostic factors (e.g., disease characteristics) on the treatment effect. The analysis of the assumptions for each Cox regression model and the overall score from each model concluded no violation against the proportional hazards (OS, *p*-value = 0.33, PFS, *p*-value = 0.48, and TTF, *p*-value = 0.65). Descriptive statistical methods were used to summarize the safety variables in both treatment arms. A Chi-square test was used to compare the response rates between the two treatment arms. All analyses were performed using the Statistical Package for Social Sciences (SPSS) version 25 (IBM Corp., Armonk, NY, USA,) and R version 3.61 (R Foundation for Statistical Computing, Vienna, Austria. URL http://www.R-project.org/).

## 5. Conclusions

The combination of cetuximab and paclitaxel/carboplatin seems to have similar efficacy and less toxicity compared to cetuximab and 5-FU/cisplatin or carboplatin, rendering it a favorable treatment option for the first-line treatment of RM-SCCHN. This less toxic treatment should be tested along with immunotherapy in future trials, in order to offer to a broader subset of patients a chance for treatment completion and durable responses. The results of our trial should be validated in a larger randomized phase 3 trial.

## Figures and Tables

**Figure 1 cancers-12-03110-f001:**
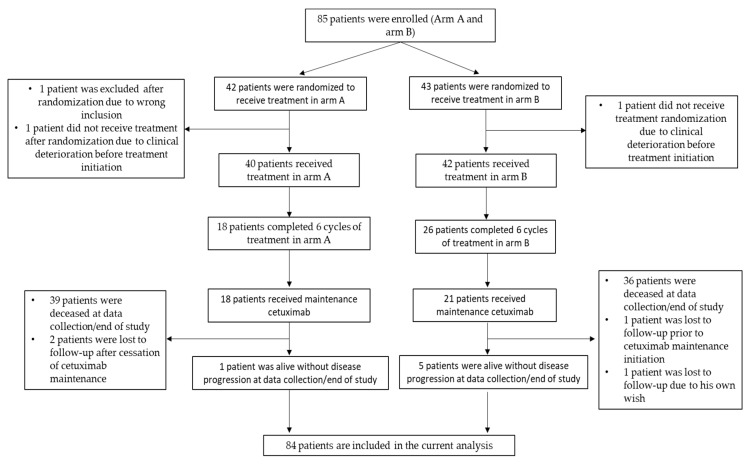
Patient disposition diagram.

**Figure 2 cancers-12-03110-f002:**
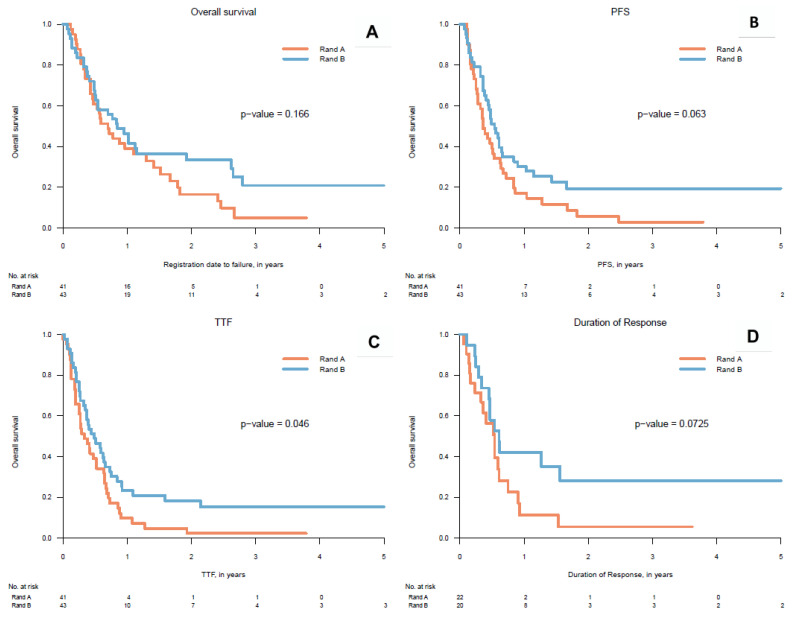
Kaplan Meier curves, rand A: Randomization Arm A, rand B: Randomization Arm B, (**A**) OS: Overall Survival, (**B**) PFS: Progression-Free Survival, (**C**) TTF: Time to Treatment Failure, (**D**) Duration Of Response.

**Table 1 cancers-12-03110-t001:** Baseline demographics and disease characteristics.

	Arm A (*N* = 42)		Arm B (*N* = 43)		*p* Value
	*n*	%	*n*	%	
**Gender**					
Male	33	78.6	26	60.5	
Female	9	21.4	17	39.5	0.07
**Localization**					
Hypopharynx	4	9.5	4	9.3	
Larynx	2	4.8	5	11.6	
Oral cavity	30	71.4	29	67.4	
Oropharynx	5	11.9	5	11.6	
Missing	1	2.4	0	0	0.734
**T stage**					
Τ0	2	4.8	0	0	
T1	3	7.1	4	9.3	
T2	17	40.5	17	39.5	
T3	12	28.6	10	23.3	
T4	7	16.7	12	27.9	0.463
**N stage**					
N0	9	21.4	17	39.5	
N1	6	14.3	6	14	
N2	23	54.8	18	41.9	
N3	1	2.4	0	0	
Nx	2	4.8	2	4.7	0.409
**M stage**					
M0	41	97.6	43	100	
M1	1	2.4	0	0	0.35
**HPV status**					
Positive	11	26.2	15	34.9	
Negative	24	57.1	27	62.8	
Missing	7	16.7	1	2.3	0.071
**Smoking status**					
Former smoker	28	66.7	27	62.8	
Non-smoker	9	21.4	6	14	
Smoker	5	11.9	9	20.9	
Not known	0	0	1	2.3	0.432
**Age**					
Mean, stdv	59.13	10.12	59.10	7.27	0.989
**Recurrent/metastatic disease**					
Local recurrence	12	36.4	21	63.6	
Metastatic	14	56	11	44	
Recurrent and metastatic	16	59.3	11	40.7	0.155
**WHO performance Status**					
0	14	34.1	15	34.9	
1	27	65.9	27	62.8	
2	0	0	1	2.3	0.61
**Prior treatment**					
Neoadjuvant chemotherapy	5	12	5	11.6	
Radiotherapy	39	93	41	95	
Chemoradiotherapy	26	62	18	42	
Surgery	20	48	23	53.5	*

T: tumor, N: node, M: metastasis, HPV: Human Papilloma Virus, stdv: standard deviation. * Each subgroup of prior treatment was tested with Fisher’s exact test, Pearson’s Chi-square, Mantel–Haenszel Chi-Square. No statistically significant difference was found between the study arms.

**Table 2 cancers-12-03110-t002:** Multivariate analysis (only corrected for gender).

	OS		PFS		TTF	
	HR	95% CI	HR	95% CI	HR	95% CI
**Gender**						
Male	1.00	ref.	1.00	ref.	1.00	ref.
Female	1.09	0.63–1.89	0.99	0.58–1.68	1.03	0.61–1.74
**Treatment**						
Arm A	1.00	ref.	1.00	ref.	1.00	ref.
Arm B	0.70	0.42–1.15	0.65	0.40–1.05	0.62	0.39–1.01

HR: Hazard Ratio, CI: Confidence Interval, OS: Overall Survival, PFS: Progression-free survival, TTF: Time to Treatment Failure, ref: reference.

**Table 3 cancers-12-03110-t003:** Adverse events ≥ grade 3.

Adverse Events	Arm A (*N* = 27)	Arm B (*N* = 24)	Total
Anemia	3	1	4
Anorexia	3	1	4
Bleeding from mouth and nose	1	1	2
Colon perforation	1	0	1
Dysphagia	2	2	4
Fatigue	6	4	10
Febrile neutropenia	6	3	9
Gingivostomatitis	1	0	1
Heart failure	1	0	1
Hypokalemia	1	2	3
Infections	9	3	12
Leukopenia	0	4	4
Mucositis oral	3	0	3
Nausea	4	2	6
Neuropathy—sensory	0	2	2
Pain	1	0	1
Rash/acne	0	1	1
Syncope	1	0	1
Thromboembolic events	3	0	3
Thrombocytopenia	2	2	4
Ulcerations on fingers and nails	1	0	1
Vomiting	1	1	2
Diarrhea	1	0	1
Hand and foot syndrome	0	1	1
Hypomagnesemia	0	1	1
Motoric neuropathy	0	1	1
Sepsis	0	1	1
Syncope	1	0	1
Weight loss	1	0	1
**Total events**	61	41	102

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
