# Peer review of "Randomized Phase II Study with Cetuximab in Combination with 5-FU and Cisplatin or Carboplatin Vs. Cetuximab in Combination with Paclitaxel and Carboplatin for Treatment of Patients with Relapsed or Metastatic Squamous Cell Carcinoma of the Head and Neck (CETMET Trial)"

_cancers, 2020, doi:10.3390/cancers12113110_

Round 1

Reviewer 1 Report

The current study details a randomized Phase II study investigating cetuximab/5FU/cisplatin or carboplatin (Arm A) vs cetuximab/paclitaxel/carboplatin (Arm B) for relapsed or metastatic squamous cell carcinoma of the head and neck. Given the relative toxicity of Arm A and poor outcomes of these patients, the authors sought to investigate whether substitution of paclitaxel for 5FU would be associated with a similar outcome but less adverse effects. The authors acknowledge that while many contemporary treatment regimens now include immunotherapy for these patients, some patients cannot tolerate the increase in treatment toxicity so this proposed regimen remains relevant. Overall I think this is an interesting question worth addressing but I have concerns with the data in its current form.

Major Comments:

  1. The study was stratified based on hypopharyngeal vs non-hypopharyngeal primary, with approximately 10% hypopharyngeal tumors in each arm. While we recognize the relative poor outcome of hypopharyngeal tumors, further breakdown by site would be beneficial, particularly proportion of oropharynx and oral cavity tumors.
  2. Expanding on point 1, there was a relative unequal distribution of HPV associated tumors in Arm B. This could confound the reported benefit of Arm B, as HPV status was not addressed statistically. In fact, the authors specifically state HPV status was well balanced (p=0.071) but gender was not (p=0.07). This is a major oversight and needs to be addressed.
  3. Arm A allows for the inclusion of either cisplatin or carboplatin. Typically carboplatin can be associated with a more favorable toxicity profile. It would be beneficial to see what portion of Arm A received cisplatin vs carboplatin. This would allow a better understanding if the more favorable toxicity of Arm B was associated with the substitution of paclitaxel for 5FU, or the cisplatin vs carboplatin.
  4. It is theorized that the improvement in TTF and trends towards better oncologic outcomes with Arm B were due to better treatment tolerance (less breaks, completed treatment, etc). It is not clear in the manuscript what portion of each Arm completed the planned number of cycles.

Minor Comments:

  1. Table 1 includes TNM at initial diagnoses. There is only one patient with de novo metastatic disease at initial diagnosis, so the vast majority had prior treatments, which need to be better described (eg surgery, how much/long prior chemo and what chemo, radiation, what was the treatment response to these prior treatments). This is also important, since results do mention PFS stratified on previous chemo and readers don’t know what previous chemo patients already received.
  2. Need censor marks on Kaplan Meier curves
  3. Need toxicity grades listed for Table 3
  4. Not everyone had PET/CT at baseline. Authors didn’t mention about whether metastatic or relapsed disease need to be biopsy-proven prior to starting treatments. With these limitations in mind, I am sure there would have been some vague cases where it could be CR vs PR vs etc. So reproducibility may be somewhat debatable.
  5. Show power analysis. Non-inferiority trial with n=80.
  6. Cox analysis’ assumptions need to be verified
  7. Need CONSORT diagram

Author Response

Dear reviewer,

We would like to thank you for dedicating your valuable time in reviewing our article. We provide you with a point- by- point response to all your comments.

Major Comments:

1.The study was stratified based on hypopharyngeal vs non-hypopharyngeal primary, with approximately 10% hypopharyngeal tumors in each arm. While we recognize the relative poor outcome of hypopharyngeal tumors, further breakdown by site would be beneficial, particularly proportion of oropharynx and oral cavity tumors.

Re: Thank you for your comment. We have added a more thorough classification in table 1 including oral cavity, oropharyngeal and laryngeal cancers. No major imbalances were observed between the 2 treatment arms.

2. Expanding on point 1, there was a relative unequal distribution of HPV associated tumors in Arm B. This could confound the reported benefit of Arm B, as HPV status was not addressed statistically. In fact, the authors specifically state HPV status was well balanced (p=0.071) but gender was not (p=0.07). This is a major oversight and needs to be addressed.

Re: This is a very important comment and we added an explanation in the text. The reason why we didn’t include HPV status in the multivariate analysis is because there was a large amount of missing data regarding HPV status in arm A compared to arm B (almost one out of five patients). This is most probably the reason why p value was 0.071. When we excluded the missing data for HPV status, we got the following HRs adjusted for HPV status:

OS HR 0.7286 95% CI 0.4339-1.223            Rand adjusted for HPV status

PFS HR 0.6034 95% CI 0.370-0.984           Rand adjusted for HPV status

TTF HR 0.6009 95% CI 0.3714-0.9722        Rand adjusted for HPV status

3. Arm A allows for the inclusion of either cisplatin or carboplatin. Typically carboplatin can be associated with a more favorable toxicity profile. It would be beneficial to see what portion of Arm A received cisplatin vs carboplatin. This would allow a better understanding if the more favorable toxicity of Arm B was associated with the substitution of paclitaxel for 5FU, or the cisplatin vs carboplatin.

Re: We had 23 (54.8%) patients that received Cisplatin and 17 (40.5%) who received Carboplatin in arm A. 2 (4.7%) patients did not receive any treatment (included in ITT analysis). It seems that the combination of 5-FU and Cisplatin (and not only Cisplatin) is responsible for the higher toxicity, since a large amount of patients received Carboplatin/5-FU in arm A. We added this information in the results and discussion.

4. It is theorized that the improvement in TTF and trends towards better oncologic outcomes with Arm B were due to better treatment tolerance (less breaks, completed treatment, etc). It is not clear in the manuscript what portion of each Arm completed the planned number of cycles.

Re: Thank you for your comment. We have clarified in the manuscript. It is 18 patients (43%) in arm A vs 26 (61%) patients in arm B. We included this information in the Consort diagram and in the results. We have commented on that in the discussion.

Minor Comments:

1. Table 1 includes TNM at initial diagnoses. There is only one patient with de novo metastatic disease at initial diagnosis, so the vast majority had prior treatments, which need to be better described (eg surgery, how much/long prior chemo and what chemo, radiation, what was the treatment response to these prior treatments). This is also important, since results do mention PFS stratified on previous chemo and readers don’t know what previous chemo patients already received.

Re: Thank you for your comment. We have added information in the results, 2.1: Patient characteristics, and added prior treatment in table 1. Each subgroup of prior treatment was tested with Fisher’s exact test, Pearson Chi-square, Mantel-Haenszel Chi-Square. No statistically significant difference was found between the study arms. All patients responded very well to treatment before study enrolment since it was given with a curative intention. The relapse was diagnosed during follow up after curative treatment.

2. Need censor marks on Kaplan Meier curves

Re: Thank you for the comment. We do agree that censor marks are useful on KM curves. However, we believe that the Number at Risk numbers are more informative than using censor marks in the Kaplan Meier curves.

3. Need toxicity grades listed for Table 3

Re: Thank you for your comment. We have added an explanation on table 3 that it is all toxicities grade 3-5 that we analysed. There is an explanation in the text about deaths (grade 5). We chose this format for the table so that it is easier to read. We believe that it will be more difficult to read if we divide grade 3 and 4 toxicities, since we do not have so many events. Please let us know if you meant something else.

4. Not everyone had PET/CT at baseline. Authors didn’t mention about whether metastatic or relapsed disease need to be biopsy-proven prior to starting treatments. With these limitations in mind, I am sure there would have been some vague cases where it could be CR vs PR vs etc. So reproducibility may be somewhat debatable.

Re: All patients had to have a biopsy proven metastatic/relapsed disease (it was one of the   inclusion criteria). This information is already written in our manuscript, section 4.1; patient eligibility. All  patients had to have at least 1 dimensionally measurable lesion either by CT scan or MRI or  physical examination according to RECIST 1.1. This was also one of the inclusion criteria.We have changed this sentence in section 4.1 so that it becomes more clear. We agree that PET CT is a very good method in order to distinguish OR vs SD vs PD. PET CT has also its limitations, especially in smaller tumours. In the methods section we explain how the radiological controls were performed. . If PET/CT was done at baseline and after cycle 2, the first tumour evaluation was replaced by PET/CT. The following tumour assessment was continued with MRI or CT. We tried to use the same method in baseline, after cycle 2 and in the first evaluation after cycle 2, so that the radiological assessment would be more objective.

5. Show power analysis. Non-inferiority trial with n=80.

Re: We have shown the power analysis in the methods section: statistical analysis 4.4: A non-inferiority log-rank test with an overall sample size of 80 subjects (40 in the reference group and 40 in the treatment group) achieves 67.7% power at a 0.100 significance level to detect an equivalence hazard ratio of 1.50 when the actual hazard ratio is an equivalence hazard ratio of 1.00 and the reference group hazard rate is 0.1238. This low power applies only to the primary endpoint.

6. Cox analysis’ assumptions need to be verified

Re: We did analyze the assumptions for each Cox Regression model, and the overall Score from each model concluded no violation against the Proportional Hazards (OS, p-value = 0.33, PFS, p-value = 0.48, and TTF, p-value = 0.65. Reference: P. Grambsch and T. Therneau (1994), Proportional hazards tests and diagnostics based on weighted residuals. Biometrika, 81, 515-26.

We added this information in the methods section; 4.5 : Statistical analysis.

7. Need CONSORT diagram

Re: We have included the CONSORT diagram in the manusript (it was in the suppl material) and changed its form (figure 1).

Reviewer 2 Report

The manuscript presents a prospective randomized phase 2 study comparing EXTREME regimen with cetuximab and paclitaxel/carboplatin in patients with recurrent/metastatic head and neck cancer. 85 patients were included and both arms were well-balanced regarding different characteristics. Primary endpoint was PFS, which was found comparable between the two arms. However, the tested regimen was significantly less toxic than the standard. The conclusion from this trial is that tested regimen may represent an equivalent alternative to standard therapy.

The manuscript is well-written, concise and the conclusions are supported with results. The only concern the issue of statistical power concerning the primary endpoint of PFS, which resulted from early termination of the study (due to poor accrual). This should be commented on in more detail.

Results (table 1): add the number (%) of recurrent, metastatic and recurrent & metastatic cases by treatment arm. If there is a difference between the arms in this respect (not stratifying factor), this factor should be statistically evaluated.

Minor issue:

Abstract: Information is missing as to why the two regimens are being compared (difference in toxicity)

Author Response

Dear reviewer,

We would like to thank you for dedicating your valuable time in reviewing our article. We provide you with a point-by-point response to all your comments.

1. The manuscript is well-written, concise and the conclusions are supported with results. The only concern the issue of statistical power concerning the primary endpoint of PFS, which resulted from early termination of the study (due to poor accrual). This should be commented on in more detail.

Re: Thank you for your comment. We have added more information on this issue in the discussion, paragraph 3.

2. Results (table 1): add the number (%) of recurrent, metastatic and recurrent & metastatic cases by treatment arm. If there is a difference between the arms in this respect (not stratifying factor), this factor should be statistically evaluated.

Re: We have included this information in table 1. P value was 0.155, therefore well balanced between the study arms.

Minor issue:

1. Abstract: Information is missing as to why the two regimens are being compared (difference in toxicity)

Re: We have added this information in the abstract.

Reviewer 3 Report

The Authors aimed to investigate whether cetuximab and paclitaxel/carboplatin can achieve similar progression free survival (PFS) with standard cetuximab and 5-FU/platinum based chemotherapy in head and neck cancer patients. The Authors demonstrated that cetuximab and paclitaxel/carboplatin was found to have similar efficacy and less toxicity compared to cetuximab and 5-FU/cisplatin or carboplatin.

The manuscript is good, experimental design and analyses are well reported. The result highly relevant is TTF was significantly better for the experimental arm B in a direct  superiority comparison with those of arm A. This is important because this finding correlates with a low toxicity for patients. Although this, the authors should clarify some points.

The Discussion should be revised and should be focused on, in particular, low toxicity following arm B treatment compared to those of A arm. Moreover, it is important also to point out the advantage and the ease of drugs administration. These points are reported but they are badly described.

The Authors should clarify in M&M that the patients do not express PDL-1. If it is not, the authors should check the correlation between therapeutic regimen and PDL1 expression. Therefore, the authors must clarify this point.

The English language must be revised by a native English speaker.

Author Response

Dear reviewer,

We would like to thank you for dedicating your valuable time in reviewing our article. We provide you with a point-by-point response to all your comments.

1. The Discussion should be revised and should be focused on, in particular, low toxicity following arm B treatment compared to those of A arm. Moreover, it is important also to point out the advantage and the ease of drugs administration. These points are reported but they are badly described.

Re: We have added more information related to toxicity at the discussion. We have added even information in the results section.  We have added information regarding the amount of patients that received all 6 planned cycles of chemotherapy in the 2 arms, as well as regarding the number of patients who received Cisplatin in arm A. We have added information regarding the ease of drug administration.

2. The Authors should clarify in M&M that the patients do not express PDL-1. If it is not, the authors should check the correlation between therapeutic regimen and PDL1 expression. Therefore, the authors must clarify this point.

Re: Thank you for the comment. We do agree that it is important today to know the PDL1 status of the tumor. However, PDL1 testing was not in clinical praxis when we conducted this trial. Therefore we don’t have any data regarding PDL1 expression. Immunotherapy was not either a part of clinical praxis when this trial was conducted. There were no published data at that time.

3. The English language must be revised by a native English speaker.

Re: A native English speaker has revised the manuscript. We have corrected accordingly.

Round 2

Reviewer 1 Report

The authors has answered all questions well.